# KLF11 deficiency enhances chemokine generation and fibrosis in murine unilateral ureteral obstruction

Silvana B. De Lorenzo[1], Alyssa M. Vrieze[2], Ruth A. Johnson[1], Karen R. Lien[1], Karl A. Nath[3], Vesna D. Garovic[3], Khashayarsha Khazaie[4], Joseph P. Grande[1]*

1 Department of Laboratory Medicine & Pathology, Mayo Clinic, Rochester, Minnesota, United States of America, 2 Department of Comparative Medicine, Mayo Clinic, Rochester, Minnesota, United States of America, 3 Division of Nephrology & Hypertension, Department of Internal Medicine, Mayo Clinic, Rochester, Minnesota, United States of America, 4 Department of Immunology, Mayo Clinic, Rochester, Minnesota, United States of America

* grande.joseph@mayo.edu

**Data Availability Statement:** All relevant data are within the manuscript and its supporting information files.

**Funding:** This work was funded by the Department of Laboratory Medicine and Pathology, Mayo Clinic

## Abstract

Progression of virtually all forms of chronic kidney disease (CKD) is associated with activation of pro-inflammatory and pro-fibrotic signaling pathways. Despite extensive research, progress in identifying therapeutic targets to arrest or slow progression of CKD has been limited by incomplete understanding of basic mechanisms underlying renal inflammation and fibrosis in CKD. Recent studies have identified Kruppel-like transcription factors that have been shown to play critical roles in renal development, homeostasis, and response to injury. Although KLF11 deficiency has been shown to increase collagen production in vitro and tissue fibrosis in other organs, no previous study has linked KLF11 to the development of CKD. We sought to test the hypothesis that KLF11 deficiency promotes CKD through upregulation of pro-inflammatory and pro-fibrogenic signaling pathways in murine unilateral ureteral obstruction (UUO), a well-established model of renal fibrosis. We found that KLF11-deficiency exacerbates renal injury in the UUO model through activation of the TGF-β/SMAD signaling pathway and through activation of several pro-inflammatory chemokine signaling pathways. Based on these considerations, we conclude that agents increase KLF11 expression may provide novel therapeutic targets to slow the progression of CKD.

## Introduction

Kruppel-like factors (KLFs) are a family of zinc-finger transcription factors that play important roles in regulation of growth and development, proliferation, and regeneration following injury [1–4]. There are over 17 members of the KLF family which can be divided into three phylogenetic groups: Group 1 (KLF 3, 8, and 12) which serve as transcriptional repressors through interaction with C terminal binding protein; Group 2 (KLF 1, 2, 4, 5, 6, 7) which are, in general, transcriptional activators; and Group 3 (KLF 9, 10, 11, 13, 14, and 16) possess an alpha helical motif that mediates binding to Sin3A and function as transcriptional repressors

College of Medicine. The funders had no role in study design, data collection and analysis, decision to publish, or preparation of the manuscript.

**Competing interests:** The authors have declared that no competing interests exist.

[5, 6]. Several KLF family members have been shown to mediate fibrosis through interaction with the TGF-β/SMAD3 signaling pathway [7, 8].

Several recent studies have implicated KLF family members in renal pathophysiology. KLFs which have been shown to mediate kidney injury include KLF 2, 4, 5, 6, and 15. Members of the KLF family have been linked to podocyte dysfunction (KLF 6, 15) and renal fibrosis (KLF 4, 5, 6, 15), at least in part through modulation of TGF-β/SMAD3 signaling [5, 9]. These KLFs are involved in a variety of critical physiologic functions, including preservation of the endothelial capillary barrier, prevention of apoptosis, regulation of mitochondrial function, and maintenance of podocyte differentiation [5].

In vitro studies have shown that KLF11 represses gene expression via Sp1-like binding sites and binds either GC-rich or CACC sequences. Although KLF11 downregulates collagen I production [10] and KLF11 deficiency enhances hepatic fibrosis [7], there have been no previous studies linking KLF11 signaling to fibrosis or inflammation in chronic renal injury, an issue that has been identified as an important area for future investigation [5]. Based on observations that KLF11 deficiency may increase inflammation in non-renal experimental systems [11], we sought to test the hypothesis that KLF11 deficiency exacerbates renal damage in the murine Unilateral Ureteral Obstruction (UUO) model of renal fibrosis [12] by promoting a pro-inflammatory and pro-fibrotic phenotype.

## Materials and methods

### Animals

Mice KLF11 knockout (KLF11 KO) [10, 13] were crossed back into a pure C57BL/6 background for more than 20 generations in our laboratory. Wild type (C57BL6/J) (WT) mice for backcrossing were purchased from the Jackson Laboratory (Bar Harbor, ME). WT mice (n = 19 UUO, n = 11 Sham) and KLF11 KO (n = 18 UUO, n = 12 Sham) mice underwent UUO or Sham surgery at 8–12 weeks of age. UUO surgery was performed by a double ligation in the right ureter at approximately halfway between the kidney and the bladder. For Sham surgery (Control) the right ureter was identified and manipulated without generating any ureteric obstruction (no ligation and no cut in the ureter). Mice were sacrificed at day 9 after ureteral ligation. All the animal procedures were approved by the Mayo Clinic Institutional Animal Care and Use Committee (IACUC) prior to conducting any experiments. These animal procedures were conducted in accordance with National Institutes of Health Guide for the Care and Use of Laboratory Animals.

### Histologic and immunohistochemical analysis

Histologic and immunohistochemical analysis was performed on right kidney tissue from mice that underwent UUO or Sham surgery. The kidney tissues were fixed in 10% neutral buffered formalin and processed using standard techniques. 5 μm histological sections were prepared and stained with hematoxylin-eosin (H&E). Renal atrophy was semi quantitatively assessed as the percentage of cortical surface area occupied by atrophic tubules, compared to the entire cortical surface area, according to methods previously established in our laboratory [14, 15]. Immunohistochemical staining was done for anti-F4/80 (1:200, BIO-RAD, Cat. No. MCA497RT), anti-KLF11 (1:1600, Novus Biological, Cat# H00008462-M03), anti-CD3 (1:100, Agilent Dako, Code Number A045), anti-CD68 (1:200, Abcam, Cat# ab125212), anti-CD163 (1:400, Abcam, Cat# ab182422), and anti-CD206 (1:800, Abcam, Cat# ab64693). Sections were stained with Sirius Red to quantitate matrix deposition. Ten random fields per kidney section were examined for each marker (F4/80, CD3, CD68, CD163, CD206 and Sirius Red) and the average of positive areas was expressed as percentage of total analyzed area. All measurements

and quantifications were performed in a blinded fashion using NIS elements BR 4.13.00 64-bit image analysis system (Nikon Instruments INC., Melville, NY) at 200 X magnification.

### Renal function assay

For exanimating renal function, blood and terminal organ harvest was performed at day 9 after the surgery. The blood levels of Albumin (Photometric, Dye Binding-Bromcresol Green), serum BUN (Urease with GLDH (coupled Enzymes)), Creatinine (CREA) (Enzymatic) and Glucose (Hexokinase G6PDG/UV (enzymatic colorimetric assay) were examined as markers of renal dysfunction.

### Real-time PCR based array analysis

Total RNA was extracted from kidney tissues (RNeasy plus Mini kit, Qiagen, Valencia, CA) and quantified by spectrophotometry (NanoDrop Technologies, Wilmington, DE). After the RNA quality was evaluated the RNAs were reverse transcribed and the first-strand cDNA was prepared from total RNA using iScript cDNA synthesis kit (Bio-Rad, Hercules, CA). The gene expression was analyzed with RT2 Profiler Inflammatory Response and Autoimmunity PCR Array (Qiagen cat. PAMM-077Z), TGF-β/BMP signaling pathway PCR Array (Qiagen cat. PAMM-035Z) and Fibrosis PCR Array (Qiagen cat. PAMM-120Z) as per manufacturer's protocol. The gene expression of the UUO samples was calculated comparing each condition with WT-Sham. Data from all conditions were normalized to mean Ct value of Gapdh and Hsp90ab1 genes [16].

### RNAseq

The raw RNA sequencing paired-end reads for the samples was processed through the Mayo Clinic RNA-Seq bioinformatics pipeline, MAP-RSeq version 3.1.4 [17]. MAP-RSeq employs the very fast, accurate and splice-aware aligner, STAR [18], to align reads to the reference human genome build hg38. Gene and exon expression quantification was performed using the Subread [19] package to obtain both raw and normalized reads (RPKM–Reads Per Kilobase per Million mapped reads. Finally, comprehensive analyses were run on the aligned reads to assess quality of the sequenced libraries. Using the raw gene counts report from MAP-RSeq, genes that are differentially expressed between the groups was assessed using the bioinformatics package DEseq. Genes found different between the groups will be reported with their magnitude of change (log2 scale).

### Statistical analysis

Data are presented as means ± SEM. One-way analysis of variance (ANOVA) or t-test were performed for comparison between groups using ANOVA followed by Student's t-test. Differences between the groups were considered statistically significant when $p \leq 0.05$. Statistical analyses and heatmaps were performed with GraphPad Prism 8 XML Project version 8.2.1 (GraphPad Software, La Jolla, CA). The Principal Component Analysis (PCA) was calculated using ClustVis 2.0 (https://biit.cs.ut.ee/clustvis/).

## Results

### Effect of KLF11 deficiency on basal gene expression of markers of inflammation, fibrosis, and TGF-β/BMP pathway

There were no significant morphologic differences identified between the WT and KLF11 KO-Sham kidneys. In particular, the glomeruli and tubules were of normal size, with no

significant interstitial fibrosis or tubular atrophy. Nevertheless, Profiler PCR Array identified significant differences in basal expression of several members of the *TGF-β* and *BMP super-families*. A summary of genes differentially expressed in KLF11 KO-Sham and WT-Sham is provided in Tables 1 and 2, S1 and S2 Tables (significant differences designated by **(a)**).

Significantly upregulated genes in KLF11 KO-Sham mice included members of the *BMP Family* (Bmp5, Bmpr1a, Bmpr1b, Bmpr2), *Cell cycle progression* (Cdkn1b), *ECM Remodeling Enzymes* (Mmp1a, Timp3), *Growth Factors* (Gdf7, Igf1), *SMAD Family* (SMAD2, SMAD7), *TGF-β Superfamily Members* (Acvr2a, Chrd, Inha, Lefty1, TGF-βr1, TGF-βr2), and *Transcription Factors* (Atf4, Id2, Tsc22d1), Bcl2, Emp1, Fst. Conversely, expression of Serpina1a (*ECM Remodeling Enzyme*) was significantly decreased in KLF11 KO-Sham mice, compared to WT-Sham mice (Table 1 and S1 Table). Expression of several members of the immune response family were downregulated in KLF11 KO-Sham mice, compared to WT-Sham mice, including the *Chemokine (CC)* (Ccl3, Ccl4, Ccl19, Ccl22), *Chemokine Receptors (CC)* (Ccr7) and Itgb2, whereas several members of the *Chemokine (CXC) ligands* (Cxcl1, Cxcl11), *Interleukin* (Il13, Il18) and Cd14 were upregulated (Table 2 and S2 Table).

## Genetic inactivation of KLF11 increases renal injury in UUO model

Other KLF family members have been linked to podocyte dysfunction and renal fibrosis but the role of KLF11 in mediating renal inflammation and fibrosis has not been established. To assess the role of KLF11 in renal injury, Unilateral Ureteral Obstruction (UUO) was performed in WT (n = 19) and KLF11 KO mice (n = 18), and kidneys were harvested 9 days after the surgery. In Sham mice, the ureter was localized and manipulated without ureteral ligation. Immunohistochemical localization of KLF11 in Sham and UUO kidney is shown in Fig 1. There was

**Table 1. Differentially expressed genes in the TGF-β/BMP pathway and fibrotic response in Sham mice compared to UUO mice.**

| | WT-Sham | KLF11 KO-Sham | WT-UUO | KLF11 KO-UUO | |
|---|---|---|---|---|---|
| **BMP Family (Bone Morphogenetic Proteins)** | | | | | |
| Bmp1 | 1±0.15 | 1.9±0.49 (a)ns | **7.9±0.99 (b)**\*\*\*\* | **13±1.2 (c)**\*\*\*\* | **(d)**\*\* |
| Bmp3 | 1±0.094 | 1.4±0.3 (a)ns | **3.1±0.43 (b)**\* | **5±0.67 (c)**\*\*\*\* | **(d)**\* |
| Bmp5 | 1±0.16 | **9.3±2.7 (a)**\* | 1.5±0.21 (b)ns | **3.7±1 (c)**\* | **(d)**\* |
| Bmp7 | 1±0.074 | 1.1±0.061 (a)ns | **0.44±0.039 (b)**\*\*\*\* | **0.58±0.05 (c)**\*\*\*\* | **(d)**\* |
| Bmpr1a | 1±0.078 | **1.6±0.2 (a)**\* | **2±0.2 (b)**\* | **3.7±0.33 (c)**\*\*\*\* | **(d)**\*\*\* |
| Bmpr1b | 1±0.18 | **1.9±0.35 (a)**\* | 1.8±0.25 (b)ns | **3.4±0.32 (c)**\*\* | **(d)**\*\* |
| Bmpr2 | 1±0.091 | **2.1±0.27 (a)**\*\* | 2.3±0.29 (b)ns | **4.5±0.55 (c)**\*\*\* | **(d)**\*\* |
| **Cell Adhesion Molecules** | | | | | |
| Itga1 | 1±0.12 | 0.94±0.082 (a)ns | **2±0.15 (b)**\*\*\*\* | **2.5±0.092 (c)**\*\*\*\* | **(d)**\* |
| Itga3 | 1±0.098 | 1.4±0.35 (a)ns | **3.4±0.3 (b)**\*\*\*\* | **5±0.46 (c)**\*\*\*\* | **(d)**\*\* |
| Itgav | 1±0.081 | 1.1±0.079 (a)ns | **2.8±0.2 (b)**\*\*\*\* | **3.4±0.2 (c)**\*\*\*\* | **(d)**\* |
| **Cell cycle progression** | | | | | |
| Cdkn1b | 1±0.092 | **1.5±0.14 (a)**\*\* | 1.8±0.16 (b)ns | **3±0.34 (c)**\*\*\* | **(d)**\*\* |
| **ECM Remodeling Enzymes** | | | | | |
| Plau | 1±0.11 | 1.4±0.22 (a)ns | 1.3±0.089 (b)ns | **2.3±0.37 (c)**\* | **(d)**\* |
| Plg | 1±0.27 | 1.8±0.57 (a)ns | 0.78±0.11 (b)ns | 2.7±0.77 (c)ns | **(d)**\* |
| Serpina1a | 1±0.3 | **0.031±0.0056 (a)**\* | 1.1±0.2 (b)ns | 0.12±0.025 (c)ns | **(d)**\*\*\* |
| Timp3 | 1±0.053 | **1.4±0.098 (a)**\*\* | **0.59±0.05 (b)**\*\*\* | **0.85±0.033 (c)**\*\*\*\* | **(d)**\*\*\* |
| Timp4 | 1±0.098 | 1.6±0.5 (a)ns | 0.68±0.071 (b)ns | 1.2±0.2 (c)ns | **(d)**\* |
| **Growth Factors** | | | | | |
| Hgf | 1±0.082 | 0.82±0.13 (a)ns | **3.9±0.27 (b)**\*\*\*\* | **4.7±0.27 (c)**\*\*\*\* | **(d)**\* |

*(Continued)*

**Table 1.** (Continued)

| | WT-Sham | KLF11 KO-Sham | WT-UUO | KLF11 KO-UUO | |
|---|---|---|---|---|---|
| **Pdgfa** | 1±0.1 | 1.2±0.18 (a)ns | **3.5±0.32 (b)**\*\*\*\* | **4.7±0.29 (c)**\*\*\*\* | **(d)**\* |
| **Pdgfb** | 1±0.078 | 1.3±0.13 (a)ns | **6±0.49 (b)**\*\*\* | **10±1.2 (c)**\*\*\*\* | **(d)**\*\* |
| **SMAD Family** | | | | | |
| **SMAD1** | 1±0.096 | 1.2±0.16 (a)ns | **3.6±0.35 (b)**\*\*\*\* | **5.5±0.45 (c)**\*\*\*\* | **(d)**\*\* |
| **SMAD2** | 1±0.03 | **1.4±0.11 (a)**\*\* | **2.4±0.16 (b)**\*\*\* | **4.3±0.34 (c)**\*\*\*\* | **(d)**\*\*\*\* |
| **SMAD3** | 1±0.088 | 1.3±0.16 (a)ns | **2.6±0.2 (b)**\*\* | **4±0.52 (c)**\*\*\*\* | **(d)**\* |
| **SMAD4** | 1±0.062 | 1.4±0.2 (a)ns | **1.8±0.12 (b)**\* | **2.6±0.24 (c)**\*\*\*\* | **(d)**\*\* |
| **SMAD5** | 1±0.088 | 1.3±0.16 (a)ns | **2.6±0.2 (b)**\*\*\* | **3.6±0.38 (c)**\*\*\*\* | **(d)**\* |
| **Smurf1** | 1±0.064 | 1.5±0.27 (a)ns | **3.4±0.37 (b)**\*\*\* | **5±0.52 (c)**\*\*\*\* | **(d)**\* |
| **TGF-β Superfamily Members** | | | | | |
| **Acvr1** | 1±0.099 | 1.5±0.22 (a)ns | **3.1±0.3 (b)**\*\* | **4±0.28 (c)**\*\*\*\* | **(d)**\* |
| **Acvr2a** | 1±0.087 | **1.7±0.3 (a)**\* | 1.6±0.19 (b)ns | **2.8±0.29 (c)**\* | **(d)**\*\* |
| **Acvrl1** | 1±0.075 | 1.2±0.17 (a)ns | **2.6±0.33 (b)**\*\* | **3.8±0.45 (c)**\*\*\*\* | **(d)**\* |
| **Lefty1** | 1±0.12 | **1.4±0.13 (a)**\* | **3.6±0.37 (b)**\*\* | **7.2±0.88 (c)**\*\*\*\* | **(d)**\*\* |
| **Ltbp1** | 1±0.12 | 1.2±0.21 (a)ns | **2.3±0.28 (b)**\* | **3.8±0.41 (c)**\*\*\*\* | **(d)**\*\* |
| **Ltbp2** | 1±0.17 | 18±11 (a)ns | 23±4.9 (b)ns | **42±6 (c)**\* | **(d)**\* |
| **Ltbp4** | 1±0.085 | 0.88±0.15 (a)ns | **2.3±0.28 (b)**\* | **3.3±0.39 (c)**\*\*\*\* | **(d)**\* |
| **TGF-β1** | 1±0.06 | 1.1±0.098 (a)ns | **7.1±0.65 (b)**\*\*\*\* | **9.3±0.58 (c)**\*\*\*\* | **(d)**\* |
| **TGF-β2** | 1±0.26 | 2.3±0.71 (a)ns | **5.2±0.63 (b)**\*\* | **12±1 (c)**\*\*\*\* | **(d)**\*\*\*\* |
| **TGF-β3** | 1±0.12 | 3.4±1.6 (a)ns | **13±2.1 (b)**\*\*\* | **20±2.2 (c)**\*\*\*\* | **(d)**\* |
| **TGF-βi** | 1±0.06 | 1±0.088 (a)ns | **8.1±0.82 (b)**\*\*\*\* | **10±0.44 (c)**\*\*\*\* | **(d)**\* |
| **TGF-βr1** | 1±0.051 | **1.6±0.19 (a)**\*\* | **3.4±0.3 (b)**\*\* | **6.7±0.71 (c)**\*\*\*\* | **(d)**\*\*\* |
| **TGF-βr2** | 1±0.079 | **1.6±0.18 (a)**\*\* | **5.7±0.55 (b)**\*\*\* | **8.7±1.1 (c)**\*\*\*\* | **(d)**\* |
| **TNF receptor superfamily** | | | | | |
| **Tnf** | 1±0.22 | 1.8±0.69 (a)ns | **23±2.5 (b)**\*\*\*\* | **33±3.6 (c)**\*\*\*\* | **(d)**\* |
| **Tnfsf10** | 1±0.065 | 2±0.4 (a)ns | 1.5±0.21 (b)ns | 3.1±0.33 (c)ns | **(d)**\*\*\* |
| **Transcription Factors** | | | | | |
| **Akt1** | 1±0.086 | 1.2±0.097 (a)ns | **2.7±0.2 (b)**\*\*\*\* | **3.4±0.17 (c)**\*\*\*\* | **(d)**\* |
| **Junb** | 1±0.12 | 1.1±0.36 (a)ns | **13±1.1 (b)**\*\*\*\* | 2.7±0.75 (c)ns | **(d)**\*\*\*\* |
| **Nfkb1** | 1±0.064 | 1±0.098 (a)ns | **4.2±0.33 (b)**\*\*\*\* | **5.3±0.24 (c)**\*\*\*\* | **(d)**\* |
| **Runx1** | 1±0.11 | 1.3±0.3 (a)ns | **29±4.1 (b)**\*\*\*\* | **54±4.9 (c)**\*\*\*\* | **(d)**\*\* |
| **Sp1** | 1±0.064 | 1.1±0.067 (a)ns | **1.5±0.11 (b)**\* | **2.3±0.17 (c)**\*\*\*\* | **(d)**\*\* |
| **Stat1** | 1±0.22 | 0.77±0.063 (a)ns | 2.3±0.2 (b)ns | **3.7±0.6 (c)**\*\*\*\* | **(d)**\* |
| **Stat6** | 1±0.1 | 1.2±0.11 (a)ns | **2.2±0.21 (b)**\*\*\* | **3.5±0.29 (c)**\*\*\*\* | **(d)**\*\* |
| **Tsc22d1** | 1±0.099 | **2.4±0.47 (a)**\* | **3.8±0.35 (b)**\* | **5.9±0.91 (c)**\*\*\* | **(d)**\* |
| **ECM Structural Constituents** | | | | | |
| **Col1a2** | 1±0.057 | 1.1±0.23 (a)ns | **17±2.4 (b)**\*\*\* | **26±3.6 (c)**\*\*\*\* | **(d)**\* |
| **Other Genes** | | | | | |
| **Emp1** | 1±0.13 | **2.7±0.79 (a)**\* | **9.2±1.4 (b)**\*\*\* | **15±1.7 (c)**\*\*\*\* | **(d)**\* |

Gene expression analysis was performed employing the pathway Detect RNA array. The table showed the differentially expressed genes by RTPCR after 9 days of Surgery Sham/UUO. Statistical significance was determined by Student's t-test. **(a)** KLF11 KO-Sham compared with WT-Sham, **(b)** WT-UUO compared with WT-Sham, **(c)** KLF11 KO-UUO compared with KLF11 KO-Sham, **(d)** KLF11 KO-UUO compared with WT-UUO. Values are means ± SEM. p values ≤0.05 were considered as significant (GraphPad Software, La Jolla, CA). Statistically significant values are highlighted in bold

\*p ≤ 0.05

\*\*p ≤ 0.01

\*\*\* p ≤ 0.001

\*\*\*\*p ≤ 0.0001, ns: not significant.

**Table 2. Differentially expressed genes of inflammatory response in KLF11 KO mice compared to WT.**

| | WT-Sham | KLF11 KO-Sham | WT-UUO | KLF11 KO-UUO | |
|---|---|---|---|---|---|
| **Complement components/regulation** | | | | | |
| C3 | 1±0.14 | 0.77±0.24 (a)ns | **44±6.1 (b)**\*\*\*\* | **70±7 (c)**\*\*\*\* | **(d)**\* |
| C4b | 1±0.4 | 0.33±0.068 (a)ns | 3.6±0.67 (b)ns | **7.7±1 (c)**\*\*\*\* | **(d)**\*\* |
| **Chemokine (CC)** | | | | | |
| Ccl2 | 1±0.14 | 0.67±0.15 (a)ns | **67±9.7 (b)**\*\*\*\* | **103±11 (c)**\*\*\*\* | **(d)**\* |
| Ccl5 | 1±0.21 | 0.71±0.17 (a)ns | 9.8±1.5 (b)ns | **30±9.6 (c)**\*\* | **(d)**\* |
| Ccl7 | 1±0.27 | 0.71±0.24 (a)ns | **156±26 (b)**\*\* | **259±39 (c)**\*\*\*\* | **(d)**\* |
| Ccl8 | 1±0.42 | 0.21±0.064 (a)ns | **21±3.7 (b)**\* | **44±7.4 (c)**\*\*\*\* | **(d)**\* |
| Ccl12 | 1±0.34 | 0.32±0.038 (a)ns | **32±3.5 (b)**\*\*\*\* | **49±7 (c)**\*\*\*\* | **(d)**\* |
| Ccl17 | 1±0.25 | 0.74±0.17 (a)ns | **34±5.3 (b)**\*\*\*\* | **49±4.3 (c)**\*\*\*\* | **(d)**\* |
| **Chemokine Receptors (CC)** | | | | | |
| Ccr2 | 1±0.22 | 0.6±0.064 (a)ns | **20±2.4 (b)**\*\*\*\* | **28±1.3 (c)**\*\*\*\* | **(d)**\* |
| **Chemokine (CXC) ligands and receptors** | | | | | |
| Cxcl1 | 1±0.24 | **2.3±0.45 (a)**\* | **117±18 (b)**\*\*\*\* | **183±14 (c)**\*\*\*\* | **(d)**\* |
| Cxcl2 | 1±0.43 | 0.58±0.093 (a)ns | **634±157 (b)**\*\*\* | 246±31 (c)ns | **(d)**\* |
| **Interleukin** | | | | | |
| Il13 | 1±0.15 | 2.3±0.69 (a)ns | **3.3±0.6 (b)**\* | **5.4±0.57 (c)**\*\* | **(d)**\* |
| Il18 | 1±0.11 | **1.4±0.099 (a)**\* | **2.5±0.28 (b)**\*\*\* | **3.7±0.33 (c)**\*\*\*\* | **(d)**\* |
| Il6 | 1±0.26 | 0.58±0.34 (a)ns | 97±34 (b)ns | 15±2.8 (c)ns | **(d)**\* |
| Il23a | 1±0.22 | 1.5±0.32 (a)ns | 3.8±0.62 (b)ns | **6.6±1.2 (c)**\*\*\* | **(d)**\* |
| **Interleukin Receptor** | | | | | |
| Il10rb | 1±0.096 | 1.2±0.14 (a)ns | **1.9±0.13 (b)**\*\*\*\* | **2.2±0.088 (c)**\*\*\*\* | **(d)**\* |
| **Toll-like receptor** | | | | | |
| Tlr1 | 1±0.13 | 0.95±0.18 (a)ns | **12±1.5 (b)**\*\*\*\* | **17±1.7 (c)**\*\*\*\* | **(d)**\* |
| Tlr9 | 1±0.15 | 0.69±0.13 (a)ns | **9±0.99 (b)**\*\*\*\* | **13±1.5 (c)**\*\*\*\* | **(d)**\* |
| **Other Immune response members** | | | | | |
| Cd14 | 1±0.069 | **1.8±0.28 (a)**\* | **24±2.9 (b)**\*\* | **59±8 (c)**\*\*\*\* | **(d)**\*\*\* |
| Cd40 | 1±0.067 | 1±0.17 (a)ns | **7±0.82 (b)**\*\*\*\* | **12±1.1 (c)**\*\*\*\* | **(d)**\*\* |
| FasL | 1±0.15 | 0.86±0.2 (a)ns | **4.2±0.51 (b)**\*\* | **7.9±1.1 (c)**\*\*\*\* | **(d)**\*\* |
| Lta | 1±0.41 | 0.34±0.078 (a)ns | **1.6±0.27 (b)ns** | **2.7±0.47 (c)**\*\* | **(d)**\* |
| Ltb | 1±0.089 | 0.76±0.21 (a)ns | **9.6±1.4 (b)**\*\* | **17±2.6 (c)**\*\*\*\* | **(d)**\* |
| Ly96 | 1±0.12 | 0.81±0.069 (a)ns | **1.6±0.11 (b)**\*\* | **1.9±0.039 (c)**\*\*\*\* | **(d)**\*\* |
| Ripk2 | 1±0.052 | 1.4±0.18 (a)ns | **2±0.13 (b)**\*\* | **2.6±0.32 (c)**\*\* | **(d)**\* |

Gene expression analysis was performed employing the pathway Detect RNA array. The table showed the differentially expressed genes by RTPCR after 9 days of Surgery UUO. Statistical significance was determined by Student's t-test. (A) KLF11 KO-UUO compared with WT-UUO. p values ≤0.05 were considered as significant (GraphPad Software, La Jolla, CA). (B) List of genes differentially expressed between WT-Sham vs WT-UUO and KLF11 KO-Sham vs KLF11 KO-UUO but not differently expressed between WT-UUO vs KLF11 KO-UUO. Statistically significant values are highlighted in bold

\*p ≤ 0.05

\*\*p ≤ 0.01

\*\*\* p ≤ 0.001

\*\*\*\*p ≤ 0.0001, ns: not significant.

minimal staining for KLF11 in Sham mice. In WT mice subjected to UUO, there was strong nuclear staining predominantly in tubular epithelial cells. Focal positive staining of glomerular podocytes was observed (Fig 1A). As expected, there was minimal staining for KLF11 in the KLF11 KO mice. A heatmap showing relative expression of other KLF family members in Sham and UUO mice is shown in Fig 1B. KLF14 and, to a lesser extent, KLF16 expression was

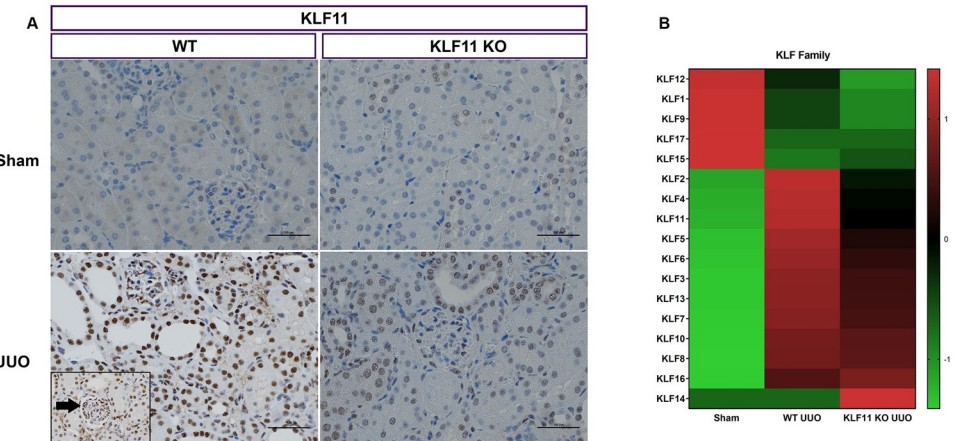

**Fig 1. Gene expression of KLF Family members in the murine unilateral ureteral obstruction (UUO) model.** (A) Representative histological images of KLF11-stained kidney sections at 400 X magnification, showing WT and KLF11 KO at 9 days after the UUO/Sham surgery. Tubular epithelial cell nuclei show strong positive staining (brown). Glomerulus (arrow, inset) shows focal positive staining within visceral epithelial cells. Scale bar represents 100 microns. (B) Heatmap. Color scale shows high and low expressions as red and green, respectively.

increased in KLF11 KO-UUO mice compared to WT-UUO mice. Several other KLF family members (KLF 2, 4, 5, 6, and 11) showed lower expression in KLF11 KO-UUO mice compared to WT-UUO mice (Fig 1B).

Tubular atrophy was semi-quantitatively assessed, in a blinded fashion, as the percentage of atrophic tubules over the entire cortical surface area, as previously described by us [14, 15]. Tubular atrophy was significantly higher in KLF11 KO mice subjected to UUO (KLF11 KO-UUO) compared to wild type mice subjected to UUO (WT-UUO) (68% vs 53%, p ≤ 0.001) (Fig 2B). No significant differences were observed in renal function parameters

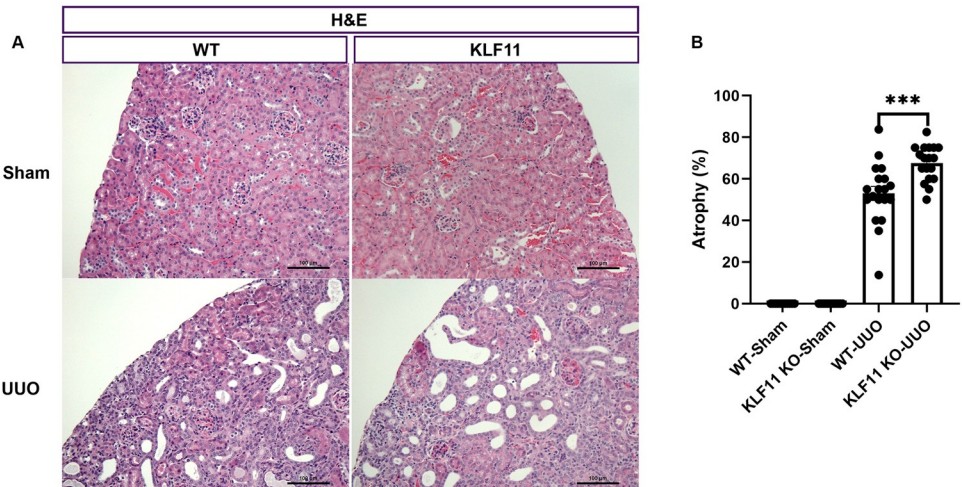

**Fig 2. KLF11 deficiency exacerbates renal damage in the murine unilateral ureteral obstruction (UUO) model.** (A) Representative histological images of H&E-stained kidney sections at 200 X magnification, showing WT and KLF11 KO at 9 days after the UUO/Sham surgery. Scale bar represents 100 microns. (B) Semiquantitative analysis of tubular atrophy. ***p ≤ 0.001. Values are means ± SEM.

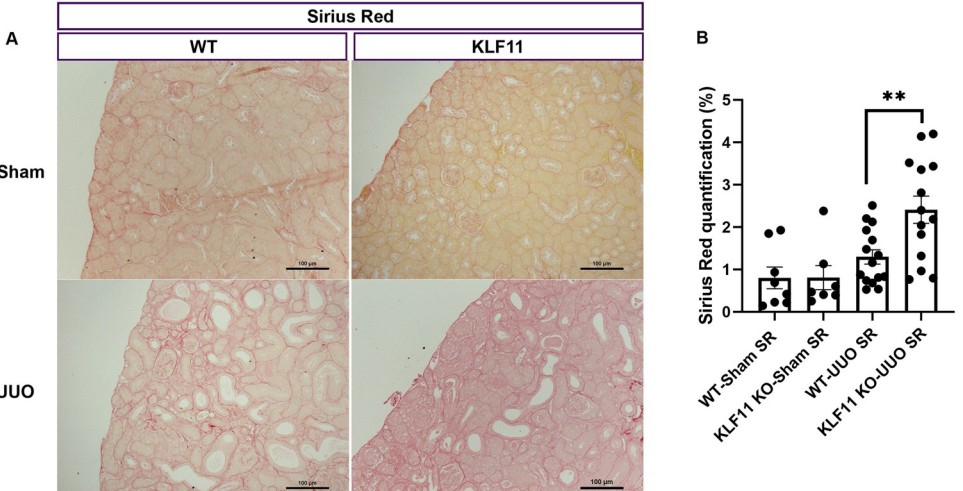

**Fig 3. Collagen deposition and expression is higher in the obstructed kidney of KLF11 KO-UUO mice compared to WT-UUO mice.** (A) representative histological images from UUO mice stained with Sirius Red (SR) at 200 X magnification, showing increased fibrosis in the obstructed kidneys of KLF11 KO-UUO comparing with WT-UUO. Scale bar represents 100 microns. (B) Quantitative analysis of the percent cortical surface area staining positively for Sirius Red (extracellular matrix). **p $\leq$ 0.01. Values are means ± SEM.

(Albumin, Blood Urea Nitrogen, Creatinine, Glucose) between WT-UUO and KLF11 KO-UUO (S3 Table).

## KLF11 deficiency increases renal fibrosis following UUO

We performed quantitative histopathologic analysis of fibrosis in KLF11 KO and WT mice subjected to UUO in Sirius Red stained slides. There were no differences in the percentage of cortical surface area staining positively for Sirius Red in the KLF11 KO-Sham (0.81±0.28) vs WT-Sham controls (0.8±0.26) (Fig 3A and 3B). Interstitial fibrosis was significantly greater in KLF11 KO-UUO (2.4±0.32) mice, compared to WT-UUO mice (1.3±0.17, p = 0.0041) (Fig 3B).

## KLF11 deletion leads to increased expression of TGF-β/BMP and other pro-fibrotic genes after UUO

We sought to identify differentially regulated genes related to the TGF-β/BMP signaling pathway and fibrosis in KLF11 KO-UUO mice compared to WT-UUO mice. Heatmaps providing a summary of differentially regulated genes in WT-Sham, KLF11 KO-Sham, WT-UUO, and KLF11 KO-UUO mice are shown in Fig 4A and 4B. Table 1 shows the classification of gene expression between: *UUO groups* where the expression of KLF11 KO-UUO genes were statistically different than WT-UUO (designated **(d)**); between *UUO* and *Sham* groups, designated **(b)** for a significant change in the expression in WT-UUO vs WT-Sham and designated **(c)** where the gene expression in KLF11 KO-UUO was significantly different than KLF11 KO-Sham; and between *Sham* groups for significant differences between the KLF11 KO-Sham and WT-Sham (designated **(a)**).

As expected from the Sirius Red results (Fig 3B), the expression of Col1a2 was significantly higher in KLF11 KO-UUO mice compared to WT-UUO mice (Table 1). Several *transcription factors* that regulate TGF-β signaling and fibrosis, including Akt1, Nfkb1, Runx1, Sp1, Stat1, Stat6 and Tsc22d1 were significantly upregulated in the KLF11 KO-UUO mice. Most members

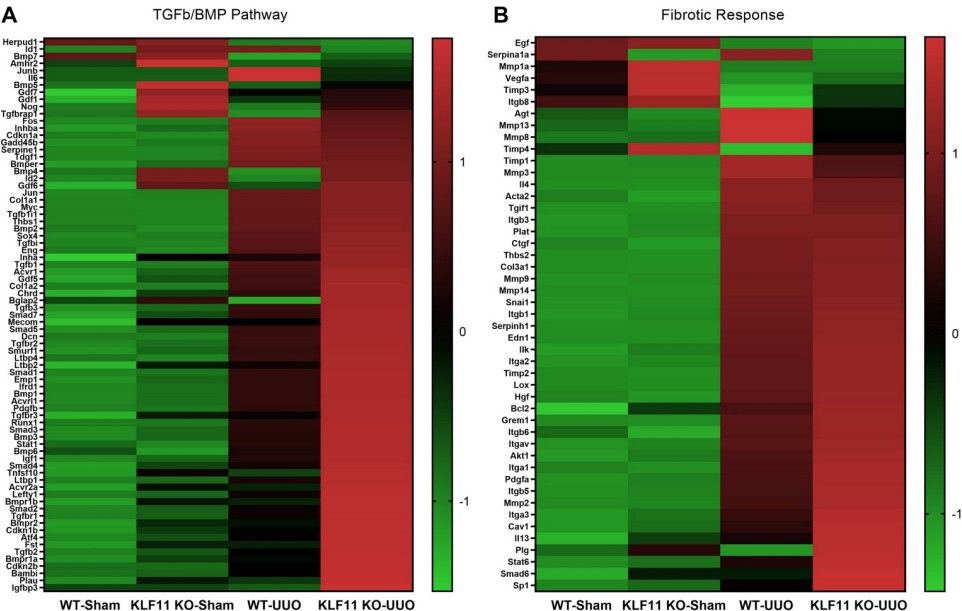

**Fig 4. Analysis of differentially expressed genes of TGF-β/BMP/Fibrotic pathways between the KLF11 KO vs WT (Sham and UUO).** Heatmap for (A) TGF-β/BMP pathway (B) Fibrotic response were generated using GraphPad Prism version 8.2.1. Color scale shows high and low expressions as red and green, respectively.

of the *BMP Family* (Bmp1, Bmp3, Bmp5, Bmp7, Bmpr1a, Bmpr1b, Bmpr2), members of the *TGF-β Superfamily Members* (TGF-β1, TGF-β2, TGF-β3, TGF-βi), *TGF-β receptors* (Acvr1, Acvr2a, Acvrl1, TGF-βr1, TGF-βr2), SMAD Family (SMADs 1–5 and Smurf1), Lefty1, Ltbp1, Ltbp2, Ltbp4, *Cell Adhesion Molecules* (Itga1, Itga3, Itgav), *Cell cycle progression* (Cdkn1b), *ECM Remodeling Enzymes* (Plau, Plg), *Growth Factors* (Hgf, Pdgfa, Pdgfb), *TNF receptor superfamily* (Tnf, Tnfsf10), and Emp1 were more highly expressed in KLF11 KO-UUO mice compared to WT-UUO mice. Conversely, Junb, Serpina1a and Timp3 expression was significantly lower in KLF11 KO-UUO mice than WT-UUO mice (Table 1).

The expression of *BMP Family* (Bmp2), *Cell Adhesion Molecules* (Itgb1, Itga2, Itgb3, Itgb5, Itgb6), *Cell cycle progression* (Cdkn1a, Gadd45b), *ECM Remodeling Enzymes* (Lox, Mmp2, Mmp3, Mmp14, Plat, Serpine1, Serpinh1, Timp1, Timp2), *Growth Factors* (Edn1, Tdgf1), *Pro-fibrotic genes* (Acta2, Ctgf), *SMAD Family* (SMAD7), *TGF-β Superfamily Members* (Cav1, Chrd, Eng, Grem1, Inhba, TGF-β1i1, Tgif1, Thbs1, Thbs2), *Transcription Factors* (Cebpb, Fos, Jun, Myc, Snai1, Sox4), *ECM Structural Constituents* (Col1a1, Col3a1) and Bcl2, were significantly higher in WT-UUO and KLF11 KO-UUO compared with the respective Shams, but there were no significant differences between KLF11 KO-UUO and WT-UUO mice (S1 Table).

Conversely, expression of Bmp7, Egf, Herpud, Mmp1a, Vegfa, Timp3 was significantly lower in the UUO (WT and KLF11 KO) compared to Sham (WT and KLF11 KO) (Table 1 and S1 Table).

## Macrophage influx increased in KLF11 KO mice compared to WT mice following UUO

**F4/80 Macrophages.** Macrophage influx was estimated as the percentage of cortical surface area staining positively for the macrophage marker F4/80 (F4/80+). F4/80+ macrophages were seen in low level in WT-Sham and KLF11 KO-Sham renal cortex. The average expression at day 9 of F4/80+ increased in the renal cortex of UUO compared with Sham. The infiltration

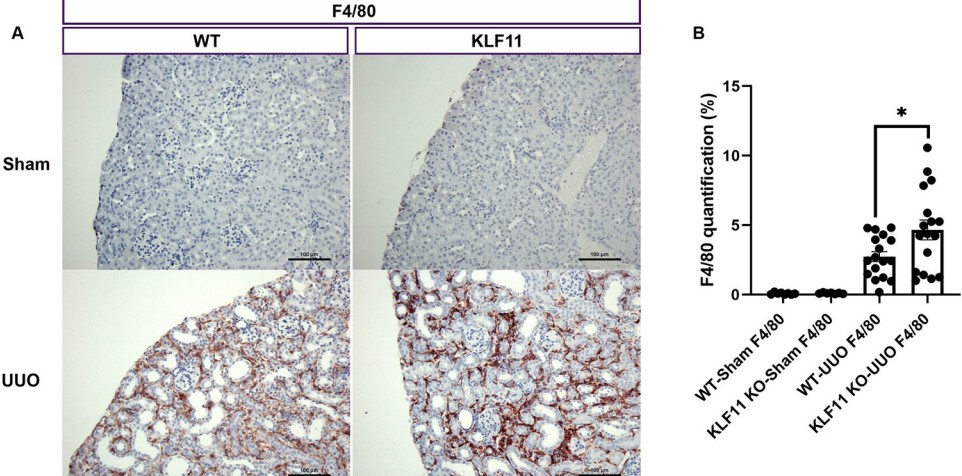

**Fig 5. KLF11 deficiency is associated with an increase in F4/80+ Macrophage influx after UUO.** Percent of cortical surface area staining positively for anti-F4/80 antibody. (A) representative histological images from Sham and UUO mice stained with anti-F4/80 antibody (F4/80) at 200 X magnification, showing increased staining in the obstructed kidneys of KLF11 KO-UUO compared toWT-UUO. Scale bar represents 100 microns. (B) Quantitative analysis of the percent cortical surface area staining positively for F4/80. $^*p \leq 0.05$; $^{**}p \leq 0.01$; $^{***}p \leq 0.001$; $^{****}p \leq 0.0001$. Values are means ± SEM.

of F4/80 increased 35 times in the WT-UUO (2.7±0.36) compared with WT-Sham (0.078 ±0.024) and more than 47 times in the KLF11 KO-UUO (4.7±0.71) compared with KLF11 KO-Sham (0.099±0.018) (Fig 5B). On day 9, the expression of F4/80+ macrophages were significantly increased in the KLF11 KO-UUO compared with WT-UUO (Fig 5B).

**CD206 macrophages.** It is recognized that "alternatively activated" M2 macrophages can mediate tissue repair as well as fibrosis. We therefore sought to determine whether UUO was associated with increased M2 macrophage influx, employing the immunohistochemical marker CD206. Only a few CD206+ macrophages were seen in the Sham controls of WT and KLF11 KO (Fig 6A). The number of CD206+ macrophages in the renal cortex was significantly increased in KLF11 KO-UUO (1.2±0.18) compared with the WT-UUO (0.69±0.099, p = 0.0302) (Fig 6B). The levels of CD206 also increased in the renal medulla but not differences were observed between WT-UUO and KLF11 KO-UUO.

We found no significant differences in macrophages staining positively for the M1 marker CD163 between WT-UUO and KLF11 KO-UUO mice. Furthermore, we found no significant differences in CD3+ T cell infiltration between the WT-UUO and

## KLF11 deletion leads to increased production of pro-Inflammatory cytokines after UUO surgery

Given that KLF11 KO-UUO mice showed increased macrophage influx compared with WT-UUO mice (Figs 5 and 6), we analyzed differentially regulated pro-inflammatory genes employing the Profile PCR array. A heatmap of differentially regulated genes in WT-Sham, KLF11 KO-Sham, WT-UUO, and KLF11 KO-UUO is shown in Fig 7A. Table 2 shows the classification of gene expression between: *UUO groups* where the expression of KLF11 KO-UUO genes were statistically different than WT-UUO (designated **(d)**); between *UUO* and *Sham* groups, designated **(b)** for a significant change in the expression in WT-UUO vs WT-Sham and designated **(c)** where the gene expression in KLF11 KO-UUO was significantly different than KLF11 KO-Sham; and between *Sham* groups for significant differences between the

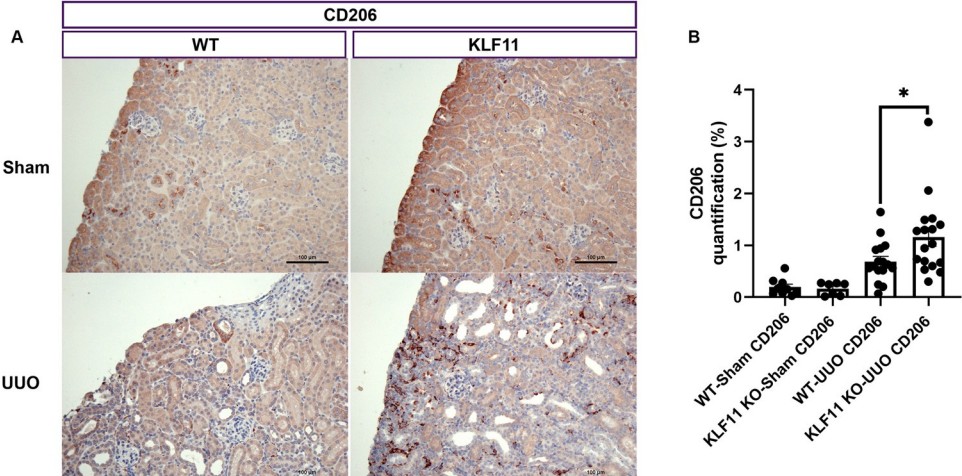

**Fig 6. KLF11 deficiency is associated with increased CD206+ macrophage influx after UUO.** Percent of cortical surface area staining positively for anti-CD206 antibody (A) representative histological images from Sham and UUO mice stained with anti-CD206 antibody (CD206) at 200 X magnification, showing increased staining in the obstructed kidneys of KLF11 KO-UUO comparedto WT-UUO. Scale bar represents 100 microns. (B) Quantitative analysis of the percent cortical surface area staining positively for CD206. *p ≤ 0.05; Values are means ± SEM.

KLF11 KO-Sham and WT-Sham (designated **(a)**). We found that expression of *Complement components/regulation* (C3, C4b), *Chemokine (CC)* (Ccl2, Ccl5, Ccl7, Ccl8, Ccl12, Ccl17), *Chemokine Receptors (CC)* (Ccr2), *Chemokine (CXC) ligands* (Cxcl1, Cxcl2), *Interleukin* (Il13, Il6, Il18, Il23a), *Interleukin Receptors* (Il10rb), *Toll-like receptor* (Tlr1, Tlr9), Cd14, Cd40, FasL, Lta, Ltb, Ly96 and Ripk2 was significantly upregulated in KLF11 KO-UUO compared to WT-UUO mice. We also found that the expression of *Complement components/regulation* (C3ar1), *Chemokine (CC)* (Ccl4, Ccl11, Ccl19, Ccl20, Ccl22), *Chemokine Receptors (CC)* (Ccr1, Ccr3, Ccr7), *Chemokine (CXC) ligands* (Cxcr4, Cxcl5), *Interleukin* (Il1b, Il7), *Interleukin Receptors* (Il1r1, Il1rap, Il1rn, Il6ra, Il23r), *Toll-like receptor* (Tlr2, Tlr3, Tlr4, Tlr5, Tlr6, Tlr7, Tirap), Bcl6, Csf1, Itgb2, Kng1, Myd88, and Nos2 were upregulated in WT-UUO and KLF11 KO-UUO compared with WT-Sham and KLF11 KO-Sham respectively. The Principal Component Analysis of the expression of genes involved in the TGF-β/BMP/Fibrotic/Inflammatory pathways indicates there are differences between the genotypes KLF11 KO-UUO and WT-UUO but not between KLF11 KO-Sham and WT-Sham (Fig 7B).

## Discussion

KLF proteins have been recognized as critical mediators of physiologic and pathophysiologic functions in many organ systems. Members of the KLF family have been linked to the pathogenesis of diabetes, obesity, inflammation, cancer, and cardiovascular disease [8, 20–22]. Although several KLF proteins have been implicated in the pathogenesis of renal disease, a potential role for KLF11 has not previously been established.

KLF KO mice are phenotypically normal, are fertile, and have normal lifespans [13]. In humans, amino acid changes in KLF11 are associated with maturity onset diabetes of the young type VII, whereas complete inactivation of this pathway by the -331-insulin mutation causes neonatal diabetes mellitus. KLF11 regulates expression of metabolic genes via an evolutionarily conserved protein interaction domain functionally disrupted in maturity onset diabetes of the young [23]. Although mutations in the KLF11 gene have been linked to maturity

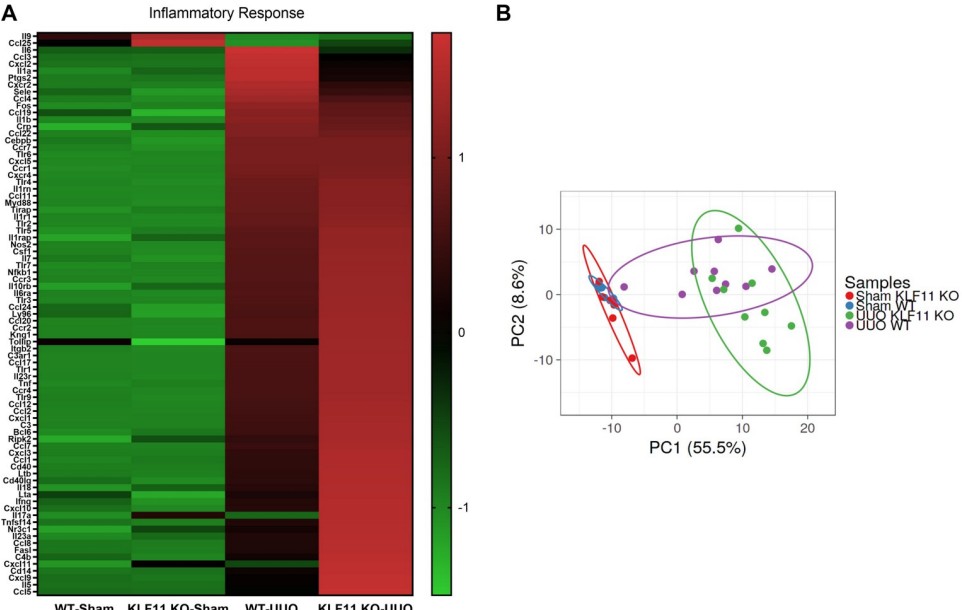

**Fig 7. Heatmap of the differential gene expression of the inflammatory response.** Gene expression of the Inflammatory response for WT-Sham, WT-UUO, KLF11 KO-Sham, KLF11 KO-UUO was measured, compared, and plotted in a Heatmap (A). Color scale shows high and low expressions as red and green, respectively. (B) PCA plot. 55.5% (PC1) and 8.6% (PC2) of the total variance (n = 33 data points). SVD with imputation is used to calculate principal components. X and Y axis show principal component 1 (PC1) and principal component 2 (PC2). Prediction ellipses are such that with probability 0.95, a new observation from the same group will fall inside the ellipse. Ellipses and shapes show clustering of the samples. Sham WT (n = 8), UUO WT (n = 9), Sham KLF11 KO (n = 6), UUO KLF11 KO (n = 9).

onset diabetes of the young type VII [9], KLF11 KO mice have decreased circulating insulin levels and increased insulin sensitivity, but do not develop overt diabetes [24, 25].

In the UUO model, ureteral obstruction leads to pressure-induced atrophy of tubular epithelial cells, leading to interstitial inflammation and fibrosis. Immunohistochemical stains for KLF11 showed relatively weak staining in WT sham mice. There was strong immunostaining for KLF11 in WT mice subjected to UUO, predominantly within tubular epithelial cells, the primary target in UUO.

We found no significant renal histopathologic alterations in the KLF11 KO Sham mice. In particular, there was no significant tubular atrophy, interstitial fibrosis, or interstitial inflammation noted. Despite the normal morphology, expression of several TGF-β/BMP family members, including TGF-βr1, TGF-βr2, and SMAD3 were higher in the KLF11 KO-Shams than WT-Shams.

We sought to test the hypothesis that KLF11 regulates inflammation and fibrosis in unilateral ureteral obstruction, a well-established model of renal fibrosis. We found that renal atrophy was more severe in KLF11 KO mice subjected to UUO, comparted to WT mice with UUO. Renal atrophy was associated with a significant influx of F4/80+ macrophages and CD206+ macrophages (a marker for M2 macrophages). There were no significant differences in the extent of CD163+ cells or CD3+ T cells between KLF11 KO-UUO mice and WT-UUO mice. Expression of pro-inflammatory cytokines was higher in KLF11 KO-UUO mice compared to WT-UUO mice (Table 2). Deposition of extracellular matrix was 1.7-fold higher in KLF11 KO-UUO mice, compared to WT-UUO mice, as assessed by Sirius Red staining. Expression of pro-fibrotic genes and genes associated with the TGF-β/BMP pathway were

significantly higher in KLF11 KO-UUO mice compared to WT-UUO mice (Table 1). Based on the strong and diffuse staining of tubular epithelial cells in WT mice subjected to UUO, we hypothesize that KLF11 deficiency may exacerbate tubular injury in response to ureteric obstruction. Support for the notion that tubular epithelial cells can initiate pro-inflammatory and pro-fibrotic signaling pathways has been obtained in other models of renal injury. In a murine model of renovascular hypertension, we found that tubular epithelial cells in the stenotic kidney strongly expressed Ccl2, a potent monocyte chemoattractant factor, within 24 hours of surgery to establish renal artery stenosis [26]. At this time point, no significant interstitial fibrosis, tubular atrophy, or interstitial inflammation was observed, indicating that, similar to ureteric obstruction, injured tubular epithelial cells can initiate proinflammatory and profibrotic pathways. We propose that KLF11 serves as a negative regulator of such pathways in injured tubular epithelial cells.

Although a role for KLF11 in the kidney has not previously been established, there is ample evidence that KLF11 plays a critical role in regulation of inflammation and fibrosis in other organs. For example, KLF11 overexpression inhibits cardiac hypertrophy and fibrosis in mice subjected to the thoracic aortic constriction model of cardiac hypertrophy [27]. In a ferric chloride induced thrombosis model, occlusion time was significantly reduced in KLF11 KO mice. Bone marrow transplantation did not correct this phenotype, indicating that vascular KLF11 inhibits arterial thrombosis [28]. Of note, KLF11 deficiency is associated with endothelial cell activation and production of pro-inflammatory molecules [11]. KLF11 directly regulates IL-6 in the brain. Knockdown of KLF11 attenuated hypoxia/regeneration injury in cardiac myocytes [29]. Knockdown of KLF11 reduced apoptosis, caspase3, and cytochrome c and mitochondrial damage. On the other hand, genetic deletion of KLF11 aggravates ischemic brain injury [30]. KLF11 KO mice were associated with progressive fibrosis in a murine model of endometriosis [31]. KLF11 is decreased in endometriosis lesions. Loss of KLF11 mediated repression of Col1a1 expression resulted in increased fibrosis [10]. KLF11 recruited SIN3A/HDAC (histone deacetylase) resulting in Col1a1 promoter deacetylation and repression. TGF-βr1 inhibitor and HAT inhibitor inhibits KLF11 signaling.

KLF11 is a TGF-β inducible immediate early gene (TIEG) and promotes the effects of TGF-β on cell growth by influencing the TGF-β-SMAD signaling pathway and regulating transcription of genes that induce either cell cycle arrest or apoptosis [32].

Of note, we found that Junb is markedly downregulated in KLF11 KO-UUO mice, compared to WT-UUO mice (Table 1). C-Jun and Junb are components of the AP-1 family of transcription factors. Overexpression of Junb inhibits SMAD-specific gene transactivation in keratinocytes and fibroblasts [33]. This finding raises the possibility that the pro-fibrotic effects of KLF11 deficiency may be, at least in part, through inhibition of Junb expression.

Of the enzymes involved in remodeling of extracellular matrix, we found that KLF11 KO-UUO mice showed significant decreases in TIMP3 and Serpina1a expression (Table 1). TIMP3 is a matrix metalloproteinase inhibiter and its deficiency is associated with renal interstitial inflammation and fibrosis [34]. Serpina1a is a serine protease inhibitor which has been shown to be downregulated in mice with diabetes [35]. Increased protease activity may lead to inflammation and tubular epithelial cell death in the UUO model.

KLF11 appears to limit inflammation, in part through binding p65 and inhibiting NF-kB signaling [11]. TGF-β is a key mediator of both fibrosis and inflammation [36, 37]. Both SMAD3/SMAD4 and KLF11 translocate to the nucleus and interact with Sp1 like and other GC-rich sequences of target genes [5, 36, 38, 39]. However, potential interactions between KLF11 and SMAD3 in regulation of matrix production have not been previously defined. As KLF11 is a TGF-β inducible gene, it is likely that tissue fibrosis results from complex interaction between KLF11 and SMAD3/TGF-β signaling. Knockdown of KLF11 attenuates hypoxia/

regeneration injury via JAK2/STAT3 signaling in H9c2 [29]. Future studies will be directed towards defining such interactions between KLF11 and TGF-β signaling pathways and how they regulate fibrosis and inflammation.

In summary, we have defined a critical role for KLF11 in regulation of both inflammation and fibrosis in unilateral ureteric obstruction, a well characterized model of renal fibrosis. In particular, KLF11 deficiency is associated with increased renal atrophy, interstitial fibrosis, and interstitial inflammation. Compared to WT-UUO, KLF11 KO-UUO mice show marked upregulation of genes associated with TGF-β signaling, fibrosis, and inflammation. Interventions to increase KLF11 expression may provide a potential therapeutic target to decrease renal inflammation and fibrosis in chronic kidney disease.

## Supporting information

**S1 Table. Differentially expressed genes of TGF-β/BMP/Fibrotic pathways between the KLF11 KO vs WT.**
(PDF)

**S2 Table. Differentially expressed genes of inflammatory response between the KLF11 KO vs WT.**
(PDF)

**S3 Table. Renal function parameters in KLF11 KO and WT mice.**
(PDF)

**S4 Table. CD3 and CD163 markers quantification in KLF11 KO and WT.**
(PDF)

## Acknowledgments

We would like to acknowledge to Chantal E. McCabe, Ph.D. from the Quantitative Health Sciences Department of Mayo Clinic for all of her assistance and the Department of Laboratory Medicine and Pathology for support of these studies.

## Author Contributions

**Conceptualization:** Silvana B. De Lorenzo, Karl A. Nath, Vesna D. Garovic, Khashayarsha Khazaie, Joseph P. Grande.

**Data curation:** Silvana B. De Lorenzo, Alyssa M. Vrieze, Ruth A. Johnson, Karen R. Lien, Joseph P. Grande.

**Formal analysis:** Silvana B. De Lorenzo, Joseph P. Grande.

**Funding acquisition:** Joseph P. Grande.

**Investigation:** Karl A. Nath, Vesna D. Garovic, Khashayarsha Khazaie, Joseph P. Grande.

**Methodology:** Joseph P. Grande.

**Project administration:** Joseph P. Grande.

**Resources:** Joseph P. Grande.

**Supervision:** Joseph P. Grande.

**Validation:** Silvana B. De Lorenzo, Joseph P. Grande.

**Visualization:** Silvana B. De Lorenzo, Joseph P. Grande.

**Writing – original draft:** Silvana B. De Lorenzo, Karl A. Nath, Vesna D. Garovic, Khashayar-sha Khazaie, Joseph P. Grande.

**Writing – review & editing:** Silvana B. De Lorenzo, Karl A. Nath, Vesna D. Garovic, Khashayarsha Khazaie, Joseph P. Grande.

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
