## [Decision Letter · Decision Letter 0]

5 Oct 2021

PONE-D-21-23487KLF11 deficiency enhances chemokine generation and activates the TGF-β/BMP fibrotic pathway in murine unilateral ureteric obstructionPLOS ONE

Dear Dr. Grande,

Thank you for submitting your manuscript to PLOS ONE. After careful consideration, we feel that it has merit but does not fully meet PLOS ONE’s publication criteria as it currently stands. Therefore, we invite you to submit a revised version of the manuscript that addresses the points raised during the review process.

We look forward to receiving your revised manuscript.

Kind regards,

Franziska Theilig

Academic Editor

PLOS ONE

Journal Requirements:

"We would like to acknowledge to Chantal E. McCabe, Ph.D. from the Quantitative Health Sciences Department of Mayo Clinic for all of her assistance and the Department of Laboratory Medicine and Pathology for support of these studies."

"This work was funded by the Department of Laboratory Medicine and Pathology, Mayo Clinic College of Medicine.

3. We note that you have included the phrase “data not shown” in your manuscript. Unfortunately, this does not meet our data sharing requirements. PLOS does not permit references to inaccessible data. We require that authors provide all relevant data within the paper, Supporting Information files, or in an acceptable, public repository. Please add a citation to support this phrase or upload the data that corresponds with these findings to a stable repository (such as Figshare or Dryad) and provide and URLs, DOIs, or accession numbers that may be used to access these data. Or, if the data are not a core part of the research being presented in your study, we ask that you remove the phrase that refers to these data

4. Please include your full ethics statement in the ‘Methods’ section of your manuscript file. In your statement, please include the full name of the IRB or ethics committee who approved or waived your study, as well as whether or not you obtained informed written or verbal consent. If consent was waived for your study, please include this information in your statement as well

Dear authors,

would you please provide renaul function parameters (Crea, BUN) and Masson Trichrome staining to judge injury?

Best regards,

Franziska Theilig

Reviewers' comments:

Reviewer's Responses to Questions

**Comments to the Author**

1. Is the manuscript technically sound, and do the data support the conclusions?

Reviewer #1: Partly

Reviewer #2: Partly

2. Has the statistical analysis been performed appropriately and rigorously? 

Reviewer #1: Yes

Reviewer #2: Yes

3. Have the authors made all data underlying the findings in their manuscript fully available?

Reviewer #1: No

Reviewer #2: Yes

4. Is the manuscript presented in an intelligible fashion and written in standard English?

Reviewer #1: Yes

Reviewer #2: Yes

5. Review Comments to the Author

Reviewer #1: In this study, the authors describe the pathological roles of KLF11 in renal fibrosis. In particular, KLF11 KO mice showed increased kidney injury and fibrosis, accompanied by upregulation of expression of pro-fibrotic and pro-inflammatory genes. There are serious concerns to be addressed.

1. The quality of H&E staining, immunohistochemical staining and heatmaps is of poor quality. For example, I can’t identify the specific marker positive cells in histological images and the gene name on the heatmaps.

2. The authors need to indicate the expression of other KLF family factors in KLF11 KO mice with or without UUO, because deletion of KLF11 may compensatory change the expression of other KLF family factors.

3. Although the authors demonstrated that macrophage infiltration and fibrosis was suppressed in the kidneys of KLF11 global KO mice, it is unclear which KLF11-expressing cells contribute to these phenotypes. Which cells are expressing KLF11 in murine normal kidneys and injured kidneys?

4. It would be interesting to indicate the association of KLF11 with kidney diseases. For example, how does endogenous KLF11 mRNA and protein expression change after UUO?

5. The author describe that the expression of CD3 or CD163-positive cells was examined in methods and results section and the expression of CD68-positive cells was examined in methods section. However, I can’t find these data in this manuscript.

Reviewer #2: 1. In Results Genetic inactivation of KLF11 increases renal injury in UUO model, authors only can get conclusion that genetic inactivation of KLF11 contribute to tubular atrophy, because tubular atrophy can not be equal to renal injury. I suggest authors detect some renal injury biomarkers in this part.

2. The title that KLF11 deficiency enhances chemokine generation and activates the TGF-β/BMP fibrotic pathway in murine unilateral ureteric obstruction is not appropriate, because KLF11 deficiency enhances chemokine generation, and increase inflammation and pro-inflammation cytokines production to result in renal damage from the text, but can not attribute to activate the TGF-β/BMP fibrotic pathway specifically.

3. From the whole study ,authors just observed a phenomenon that KLF11 deficiency results in renal injury, accompanied with chemokine generation enhanced and inflammation and pro-inflammation cytokines production increased, but no specific mechanism.

6. PLOS authors have the option to publish the peer review history of their article (what does this mean?). If published, this will include your full peer review and any attached files.

Reviewer #1: No

Reviewer #2: No

---

## [Author Response · Author response to Decision Letter 0]

2 Dec 2021

PONE-D-21-23487

KLF11 deficiency enhances chemokine generation and activates the TGF-β/BMP fibrotic pathway in murine unilateral ureteric obstruction

PLOS ONE

We would like to thank the reviewers for their thoughtful comments provided in review of our manuscript, “KLF11 deficiency enhances chemokine generation and activates the TGF-β/BMP fibrotic pathway in murine unilateral ureteric obstruction”. We have conducted additional experiments and have made a number of clarifications in response to the constructive comments provided. A summary of the changes is provided below:

Funding information should not be included in the Acknowledgements Section of the Manuscript.

 We have removed the statement indicating that the Department of Laboratory Medicine and Pathology provided institutional support for these studies in the acknowledgements section. As suggested, we wish to update the Funding Statement to indicate:

"This work was funded by the Department of Laboratory Medicine and Pathology, Mayo Clinic College of Medicine. The funders had no role in study design, data collection and analysis, decision to publish, or preparation of the manuscript."

We note that you have included the phrase “data not shown” in your manuscript. Unfortunately, this does not meet our data sharing requirements. PLOS does not permit references to inaccessible data. We require that authors provide all relevant data within the paper, Supporting Information files, or in an acceptable, public repository. Please add a citation to support this phrase or upload the data that corresponds with these findings to a stable repository (such as Figshare or Dryad) and provide and URLs, DOIs, or accession numbers that may be used to access these data. Or, if the data are not a core part of the research being presented in your study, we ask that you remove the phrase that refers to these data

We have provided all data in supporting information files. 

Please include your full ethics statement in the ‘Methods’ section of your manuscript file. In your statement, please include the full name of the IRB or ethics committee who approved or waived your study, as well as whether or not you obtained informed written or verbal consent. If consent was waived for your study, please include this information in your statement as well

This study does not involve human subjects. We have included the statement that “all animal procedures were approved by the Mayo Clinic Institutional Animal Care and Use Committee (IACUC) prior to conducting any experiments. These animal procedures were conducted in accordance with then National Institutes of Health Guide for the Care and Use of Laboratory Animals”. 

Dear authors,

would you please provide renal function parameters (Crea, BUN) and Masson Trichrome staining to judge injury?

In the unilateral ureteral obstruction model, the contralateral kidney provides normal renal function. Therefore, changes in serum creatinine are not anticipated. We have done additional studies to measure BUN and albuminuria and have provided this information in a supplementary table. We have used Sirius Red staining to provide a quantitative assessment of extracellular matrix deposition. Sirius Red staining provides a more accurate assessment of matrix deposition than trichrome staining, as edema and non-collagenous components can stain blue with a trichrome stain. We have compared Sirius Red staining with trichrome staining in our previously published study [1].

Reviewer #1: In this study, the authors describe the pathological roles of KLF11 in renal fibrosis. In particular, KLF11 KO mice showed increased kidney injury and fibrosis, accompanied by upregulation of expression of pro-fibrotic and pro-inflammatory genes. There are serious concerns to be addressed.

1. The quality of H&E staining, immunohistochemical staining and heatmaps is of poor quality. For example, I can’t identify the specific marker positive cells in histological images and the gene name on the heatmaps. 

We have revised the figures to provide higher resolution images of the immunohistochemical staining and have enlarged the annotations to the heatmaps. 

2. The authors need to indicate the expression of other KLF family factors in KLF11 KO mice with or without UUO, because deletion of KLF11 may compensatory change the expression of other KLF family factors.

We performed additional studies employing RNASeq to assess other KLF family members in KLF11 KO mice with or without UUO. Our data are summarized in a heatmap showing relative expression of the KLF family members in sham and UUO mice (Fig 1 B). We see largest induction of KLF14 and KLF 16 in KLF11 KO UUO compared to WT UUO; other KLF members showed modest changes according to genotype.

3. Although the authors demonstrated that macrophage infiltration and fibrosis was suppressed in the kidneys of KLF11 global KO mice, it is unclear which KLF11-expressing cells contribute to these phenotypes. Which cells are expressing KLF11 in murine normal kidneys and injured kidneys?

We performed additional immunohistochemical studies to identify KLF11 staining cells. We found nuclear staining for KLF11, primarily in proximal and distal tubular epithelial cells, with focal glomerular staining of visceral and parietal epithelial cells. We observed stronger staining in WT mice subjected to UUO, compared to sham, consistent with our RNASeq data showing induction of KLF11 in WT UUO mice compared to sham. 

4. It would be interesting to indicate the association of KLF11 with kidney diseases. For example, how does endogenous KLF11 mRNA and protein expression change after UUO?

Although other KLF family members have been implicated in human and experimental kidney disease, to the best of our knowledge, this is the first report associating KLF11 with kidney disease. At the RNA and protein level, as assessed by RNASeq and immunohistochemistry, respectively, we do demonstrate that, in WT mice, KLF11 is induced with UUO. 

5. The author describe that the expression of CD3 or CD163-positive cells was examined in methods and results section and the expression of CD68-positive cells was examined in methods section. However, I can’t find these data in this manuscript.

We have provided these data in a supplemental table. 

Reviewer #2: 1. In Results Genetic inactivation of KLF11 increases renal injury in UUO model, authors only can get conclusion that genetic inactivation of KLF11 contribute to tubular atrophy, because tubular atrophy can not be equal to renal injury. I suggest authors detect some renal injury biomarkers in this part.

Tubular atrophy is a well-recognized feature of chronic renal injury. For example, ct scores are an integral component of chronic tubular injury scoring according to the Banff Classification of transplant pathology. [2]. We have extensively employed tubular atrophy as an index of chronic tubular injury in our previous publications involving both human and experimental studies [3] [4] [5] [6]. As expected with a unilateral injury model, we did not detect significant differences in serum creatinine among the experimental groups. 

2. The title that KLF11 deficiency enhances chemokine generation and activates the TGF-β/BMP fibrotic pathway in murine unilateral ureteric obstruction is not appropriate, because KLF11 deficiency enhances chemokine generation, and increase inflammation and pro-inflammation cytokines production to result in renal damage from the text, but can not attribute to activate the TGF-β/BMP fibrotic pathway specifically.

KLF11 was originally described as a Transforming Growth Factor Beta inducible immediate early gene (TIEG) [7]. Based on this consideration, it was reasonable to focus on the TGF-beta/SMAD pathway; we have shown significant perturbations in this pathway in KLF11 deficient mice. Nevertheless, we have removed “activation of the TGF-beta/BMP fibrotic pathway” from the title. 

3. From the whole study, authors just observed a phenomenon that KLF11 deficiency results in renal injury, accompanied with chemokine generation enhanced and inflammation and pro-inflammation cytokines production increased, but no specific mechanism.

To our knowledge, this is the first report linking KLF11 deficiency to renal injury. Characterization of differentially regulated pathways is an important first step in the development of a mechanistic hypothesis whereby KLF11 deficiency results in renal injury. Based on initial characterization of KLF11 as a TIEG, we hypothesized that the renal damage was associated with alterations in TGF-beta signaling. Futures studies will focus on a mechanism whereby KLF11 deficiency promotes kidney injury, with a focus on the TGF-beta pathway.

1. Diaz Encarnacion M, Griffin M, Slezak J, Bergstralh E, Stegall M, Velosa J, et al. Correlation of quantitative digital image analysis with glomerular filtration rate in CAN. Am J Transplantation. 2004;4(2):248-56. PubMed PMID: 14974947.

2. Racusen LC, Solez K, Colvin RB, Bonsib SM, Castro MC, Cavallo T, et al. The Banff 97 working classification of renal allograft pathology. Kidney International. 1999;55(2):713-23.

3. Helgeson ES, Mannon R, Grande J, Gaston RS, Cecka MJ, Kasiske BL, et al. i-IFTA and chronic active T cell-mediated rejection: A tale of 2 (DeKAF) cohorts. Am J Transplant. 2021;21(5):1866-77. Epub 2020/10/15. doi: 10.1111/ajt.16352. PubMed PMID: 33052625.

4. Grande JP, Helgeson ES, Matas AJ. Correlation of Glomerular Size With Donor-Recipient Factors and With Response to Injury. Transplantation. 2021;105(11):2451-60. Epub 2020/12/05. doi: 10.1097/TP.0000000000003570. PubMed PMID: 33273317; PubMed Central PMCID: PMCPMC8166916.

5. Kashyap S, Osman M, Ferguson CM, Nath MC, Roy B, Lien KR, et al. Ccl2 deficiency protects against chronic renal injury in murine renovascular hypertension. Scientific Reports. 2018;8(1):8598.

6. Kashyap S, Boyilla R, Zaia PJ, Ghossan R, Nath KA, Textor SC, et al. Development of renal atrophy in murine 2 kidney 1 clip hypertension is strain independent. Res Vet Sci. 2016;107:171-7.

7. Lin L, Mahner S, Jeschke U, Hester A. The Distinct Roles of Transcriptional Factor KLF11 in Normal Cell Growth Regulation and Cancer as a Mediator of TGF-beta Signaling Pathway. Int J Mol Sci. 2020;21(8). Epub 2020/04/26. doi: 10.3390/ijms21082928. PubMed PMID: 32331236; PubMed Central PMCID: PMCPMC7215894.

---

## [Decision Letter · Decision Letter 1]

24 Jan 2022

PONE-D-21-23487R1KLF11 deficiency enhances chemokine generation and fibrosis in murine unilateral ureteric obstructionPLOS ONE

Dear Dr. Grande,

Thank you for submitting your manuscript to PLOS ONE. After careful consideration, we feel that it has merit but does not fully meet PLOS ONE’s publication criteria as it currently stands. Therefore, we invite you to submit a revised version of the manuscript that addresses the points raised during the review process.

We look forward to receiving your revised manuscript.

Kind regards,

Franziska Theilig

Academic Editor

PLOS ONE

Journal Requirements:

Reviewers' comments:

Reviewer's Responses to Questions

**Comments to the Author**

1. If the authors have adequately addressed your comments raised in a previous round of review and you feel that this manuscript is now acceptable for publication, you may indicate that here to bypass the “Comments to the Author” section, enter your conflict of interest statement in the “Confidential to Editor” section, and submit your "Accept" recommendation.

Reviewer #1: All comments have been addressed

Reviewer #3: (No Response)

2. Is the manuscript technically sound, and do the data support the conclusions?

Reviewer #1: Yes

Reviewer #3: Partly

3. Has the statistical analysis been performed appropriately and rigorously? 

Reviewer #1: Yes

Reviewer #3: Yes

4. Have the authors made all data underlying the findings in their manuscript fully available?

Reviewer #1: Yes

Reviewer #3: Yes

5. Is the manuscript presented in an intelligible fashion and written in standard English?

Reviewer #1: Yes

Reviewer #3: No

6. Review Comments to the Author

Reviewer #1: In this revision, the authors have satisfactorily addressed my previous comments, and I do not have any additional comments.

Reviewer #3: In this revised manuscript, Grande et al. present data demonstrating that the absence of KLF11 expression exacerbates renal injury following unilateral ureteral obstruction (UUO). The manuscript is well-organized and easy to read. Its findings are also novel, as the effect of KLF11 expression in the kidney has never been investigated following UUO before. A few issues must be addressed, prior to this reviewer's endorsement of the manuscript for publication.

Major Issues

1. (Figure 1) Expression of KLF11 by the specific cells described in lines 177-178 of the manuscript (tubular epithelial cells, podocytes, and infiltrating mononuclear cells) is difficult to discern in the light micrograph of the WT-UUO kidney section. Is the KLF11 shown in brown? Are glomeruli shown in these micrographs? I would recommend that the authors either provide additional micrographs that more clearly portray the expression and/or use arrows or high-magnification insets to demonstrate the expression by these different cells more clearly. Also, more specific details are required in the figure legend.

2. (Table 2) Was the expression of anti-inflammatory or regulatory genes (e.g., FoxP3, IL-10) profiled as well? Did the expression of these genes vary between groups?

3. This reviewer suggests expanding the Discussion section to include details about how the authors believe KLF11 expression by various cellular subsets (podocytes, tubular epithelial cells, and monocytes) could reduce renal injury following UUO.

Minor Issues:

1. (line 3) Suggest changing title from "ureteric" to "ureteral"

2. (lines 293-295) Suggest changing interpretation of the immunohistochemistry images from "expression" to "infiltration" of macrophages.

3. (lines 306-307) Suggest defining CD163 as a marker. Is this meant to detect pro-inflammatory M1 macrophages?

4. Suggest revision for English in the manuscript (especially the revised sections), as there are some errors in grammar and syntax.

7. PLOS authors have the option to publish the peer review history of their article (what does this mean?). If published, this will include your full peer review and any attached files.

Reviewer #1: No

Reviewer #3: No

---

## [Author Response · Author response to Decision Letter 1]

23 Feb 2022

Reviewer #1: In this revision, the authors have satisfactorily addressed my previous comments, and I do not have any additional comments.

We wish to thank this reviewer for the constructive comments that have improved the manuscript. 

Reviewer #3:

1. (Figure 1) Expression of KLF11 by the specific cells described in lines 177-178 of the manuscript (tubular epithelial cells, podocytes, and infiltrating mononuclear cells) is difficult to discern in the light micrograph of the WT-UUO kidney section. Is the KLF11 shown in brown? Are glomeruli shown in these micrographs? I would recommend that the authors either provide additional micrographs that more clearly portray the expression and/or use arrows or high-magnification insets to demonstrate the expression by these different cells more clearly. Also, more specific details are required in the figure legend.

The images provided are immunostains for KLF11. The WT-UUO section shows strong nuclear staining for KLF11 (dark brown stain). There is a glomerulus in the micrograph, which we have now indicated. 

In the ureteric obstruction model, ligation of the ureter produces pressure induced dilation of tubules, which lead to tubular epithelial cell injury, interstitial fibrosis, tubular atrophy, and interstitial inflammation. Other compartments of the kidney (glomeruli, vasculature, etc.) are secondarily affected (see response to point 3, below). 

2. (Table 2) Was the expression of anti-inflammatory or regulatory genes (e.g., FoxP3, IL-10) profiled as well? Did the expression of these genes vary between groups?

We did not find significant differences in anti-inflammatory or regulatory genes between WT and KO UUO groups. 

3. This reviewer suggests expanding the Discussion section to include details about how the authors believe KLF11 expression by various cellular subsets (podocytes, tubular epithelial cells, and monocytes) could reduce renal injury following UUO.

We have expanded the discussion and added a reference to support our hypothesis that injury to KLF11 deficient tubular epithelial cells exacerbates pro-inflammatory and pro-fibrotic signaling pathways triggered by ureteric obstruction, a finding supported by our RNA expression data obtained from renal cortex. 

In the ureteric obstruction model, the primary/initial target of injury is the tubular epithelial cell (see response to #1, above). In WT-UUO mice, we observed a strong induction of KLF11 staining, primarily within tubular epithelial cells. Based on this observation, we believe that injury to the tubular epithelial cell initiates a cascade of events leading to interstitial inflammation, interstitial fibrosis, and tubular atrophy. We propose that KLF11 deficiency exacerbates tubular injury through upregulation of pro-inflammatory and pro-fibrotic signaling pathways, as outlined in the results section. 

Support for the notion that tubular epithelial cells can orchestrate pro-inflammatory and pro-fibrotic signaling pathways was obtained in our previous work using a murine model of renovascular hypertension initiated by placement of a cuff on the right renal artery, restricting blood flow by 75-80% (reference 26, added to address this concern). In this study, we subjected Ccl2-RFP reporter mice to renal artery stenosis surgery. We found a strong induction of Ccl2 (a potent chemoattractant for mononuclear cells) within tubular epithelial cells within 24 hours of renal artery stenosis surgery, a time point at which there was no significant tubular atrophy, interstitial fibrosis, or interstitial inflammation (see reference 26, added to this revision). Based on these considerations, we believe that the tubular epithelial cells—which are the target of UUO and show a striking increase in immunostaining for KLF11 following UUO—are responsible for the recruitment of inflammatory cells and initiate pro-fibrotic signaling pathways.

Minor Issues:

1. (line 3) Suggest changing title from "ureteric" to "ureteral"

Thank you for the suggestion. We have made the recommended change in line 3. 

2. (lines 293-295) Suggest changing interpretation of the immunohistochemistry images from "expression" to "infiltration" of macrophages.

We have made the suggested change in line 292.

3. (lines 306-307) Suggest defining CD163 as a marker. Is this meant to detect pro-inflammatory M1 macrophages?

CD163 was used to detect pro-inflammatory M1 macrophages. This has been clarified in line 307-308.

4. Suggest revision for English in the manuscript (especially the revised sections), as there are some errors in grammar and syntax.

We have corrected errors in grammar and syntax.

---

## [Decision Letter · Decision Letter 2]

22 Mar 2022

KLF11 deficiency enhances chemokine generation and fibrosis in murine unilateral ureteral obstruction

PONE-D-21-23487R2

Dear Dr. Grande,

We’re pleased to inform you that your manuscript has been judged scientifically suitable for publication and will be formally accepted for publication once it meets all outstanding technical requirements.

Kind regards,

Franziska Theilig

Academic Editor

PLOS ONE

Additional Editor Comments (optional):

Reviewers' comments:

Reviewer's Responses to Questions

**Comments to the Author**

1. If the authors have adequately addressed your comments raised in a previous round of review and you feel that this manuscript is now acceptable for publication, you may indicate that here to bypass the “Comments to the Author” section, enter your conflict of interest statement in the “Confidential to Editor” section, and submit your "Accept" recommendation.

Reviewer #3: (No Response)

2. Is the manuscript technically sound, and do the data support the conclusions?

Reviewer #3: Yes

3. Has the statistical analysis been performed appropriately and rigorously? 

Reviewer #3: Yes

4. Have the authors made all data underlying the findings in their manuscript fully available?

Reviewer #3: (No Response)

5. Is the manuscript presented in an intelligible fashion and written in standard English?

Reviewer #3: Yes

6. Review Comments to the Author

Reviewer #3: This reviewer is grateful for the diligence of the authors in addressing each comment directly and completely. I have no remaining concerns.

7. PLOS authors have the option to publish the peer review history of their article (what does this mean?). If published, this will include your full peer review and any attached files.

Reviewer #3: No

---

## [Editor Report · Acceptance letter]

31 Mar 2022

PONE-D-21-23487R2 

KLF11 deficiency enhances chemokine generation and fibrosis in murine unilateral ureteral obstruction 

Dear Dr. Grande:

I'm pleased to inform you that your manuscript has been deemed suitable for publication in PLOS ONE. Congratulations! Your manuscript is now with our production department. 

Kind regards, 

on behalf of

Dr. Franziska Theilig 

Academic Editor

PLOS ONE